# Indocyanine Green (ICG)-Guided Onlay Preputial Island Flap Urethroplasty for the Single-Stage Repair of Hypospadias in Children: A Case Report

**DOI:** 10.3390/ijerph20136246

**Published:** 2023-06-28

**Authors:** Irene Paraboschi, Michele Gnech, Dario Guido Minoli, Erika Adalgisa De Marco, Giovanni Parente, Guglielmo Mantica, Gianantonio Manzoni, Alfredo Berrettini

**Affiliations:** 1Department of Pediatric Urology, Fondazione IRCCS Ca’ Granda Ospedale Maggiore Policlinico, 20122 Milan, Italy; 2Department of Pediatric Surgery, ASST Papa Giovanni XXIII Hospital, 24127 Bergamo, Italy; 3Department of Urology, IRCCS Ospedale Policlinico San Martino, 16132 Genoa, Italy

**Keywords:** hypospadias, onlay preputial island flap, urethroplasty, fluorescence-guided surgery, indocyanine green, children

## Abstract

First described by Duckett in 1981, and initially employed for the surgical correction of mid-penile hypospadias, the onlay preputial island flap urethroplasty has progressively gained increasing popularity, extending its indication to proximal forms. However, with the complexity of the penile anomaly, the rate of postoperative complications related to poor tissue perfusion (including skin and glans dehiscence, urethral stenoses, and fistulas) has also increased. Conventionally, the visual assessment of the onlay preputial island flap is the only option available to establish the appropriate tissue vascularization during surgery. To this end, we have first introduced the EleVision IR system (Medtronic Ltd., Hong Kong, China) to assess the vascular perfusion of the preputial island flap in a 13-month-old boy undergoing the onlay urethroplasty for the surgical correction of a mid-shaft hypospadias. This was possible 80 s after the intravenous injection of indocyanine green (ICG, 0.15 mg/kg, Diagnostic Green GmbH, Munich, Germany). ICG-based laser angiography helped define the proximal resection margin of the preputial flap, and proved to be safe, effective, and easy to employ. This innovative intraoperative imaging modality can be considered a useful adjunct for tissue perfusion evaluation and intraoperative decision-making during the onlay preputial island flap urethroplasty in children.

## 1. Introduction

First described by Duckett in 1981, and initially employed for the surgical correction of mid and distal penile hypospadias, the onlay preputial island flap urethroplasty has progressively gained more popularity, extending its indication to proximal forms [1,2,3,4,5,6].

This surgical technique has proved to be particularly useful in children with a well-developed urethral plate, who exhibit little or no curvature after the release of the ventral penile chordees [1,2,3,4,5,6].

It involves the mobilization of the entire inner aspect of the prepuce, which is then brought to the ventrum of the penis on a vascular pedicle of adequate length.

The onlay preputial island flap is then fitted for the urethral plate, by excising the excess of the inner prepuce, and sutured onto it as an onlay with running sutures on each side for the urethroplasty on a catheter of adequate size. The glans wings are finally closed to form an apical urinary meatus [1,2,3,4,5,6].

Worth mentioning is that, with the complexity of the penile anomaly and the more proximal position of the external urethral meatus, the rate of postoperative complications related to poor vascular supply (including glans dehiscence, urethral stenosis, and fistula) has also increased, reaching up to 30–68% of cases [3,4,6]. 

In fact, a sufficient blood supply of the pedicled skin graft is one of the most important and critical factors for a successful surgical repair.

Conventionally, the visual assessment of the preputial island flap is the only option available to determine if the vascular perfusion of the preputial island flap is good, and to assess its viability during surgery.

More recently, however, fluorescence-guided surgery (FGS) has emerged as a very promising intraoperative imaging modality that can help surgeons assess the vascular perfusion of the tissues they operate on, in real-time, by administering near-infrared (NIR) fluorescent dyes [7,8,9,10]. 

Amongst them, indocyanine green (ICG) has largely proved to be a safe and effective fluorophore that can be administered prior to or during surgery, to guide procedures in which the tissue vascular perfusion is critical for the success of the surgical outcomes [7,8,9,10]. 

Already described for other urogenital procedures in pediatric urology and plastic surgery, the usefulness of ICG-based laser angiography in hypospadias repair has never been described before [7,8,9,10].

The aim of this case report is, therefore, to present the first step-by-step protocol to perform ICG-based FGS during the surgical repair of a male infant with a mid-shaft form of hypospadias. A new tool capable of assessing the vascular perfusion of the onlay preputial island flap urethroplasty was safely adopted, allowing surgeons to reach successful and lasting surgical outcomes. 

## 2. Detailed Case Description

A 13-month-old boy referred to our tertiary referral hospital for mid-shaft hypospadias was scheduled for the onlay preputial island flap urethroplasty (Figure 1a). 

In this specific case, a preoperative course of androgens was not required, because the urethral plate, glans, and penile shaft were all well-developed.

At the time of the surgical procedure, 0.15 mg/kg of ICG (Diagnostic Green GmbH, Munich, Germany) was intravenously administered and the EleVision IR system (Medtronic Ltd.) was adopted intraoperatively to assess the vascular perfusion of the preputial island flap in real-time (Appendix A).

As the standard onlay preputial island flap urethroplasty, the procedure began with the demarcation of the urethral plate and the catheterization of the mid-shaft meatus (Figure 2a). Following the application of a tourniquet at the base of the penis for maintaining a bloodless field, the penis was degloved, preserving the urethral plate. An artificial erection test was then performed, which showed no ventral penile curvature (Figure 2b). After measuring the length of the urethra to be reconstructed, a transverse preputial island flap of suitable length (i.e., 35 mm) was then prepared (Figure 2c). At this point, 0.15 mg/kg of ICG (Diagnostic Green GmbH, Munich, Germany) was administered, and the EleVision IR system (Medtronic Ltd.) was incorporated to help assess the perfusion of the preputial island flap in real-time. The aim of their employment during surgery was to reduce the complications associated with a poorly vascularized preputial flap (Figure 3). Favorable fluorescence was obtained 80 s after the injection, confirming the proper vascularization of the onlay preputial island flap elevated on its pedicle, especially on its proximal left side (Figure 3a,b). Hence, the ICG-based navigation system applied to the surgical field helped define the proximal resection margin, and guided the excision of the distal part, significantly impacting the intraoperative decision-making (Figure 3c,d).

The preputial island flap was then placed as an onlay over an 8 Ch catheter, to create the neo-urethra. Again, the perfusion of the ventral part of the neo-urethra was assessed by using the EleVision IR system (Medtronic Ltd.), and proved to be adequate (Figure 3e,f). A single-layer glanduloplasty and skin closure followed, as in standard onlay repairs (Figure 1b). 

No ICG-related side effects were experienced during surgery, and the postoperative course was also favorable. The patient was discharged 2 days after surgery, and the 8 Ch catheter was removed on postoperative day (POD) 7. At the 11-month follow-up, the esthetic and functional outcomes proved to be excellent, confirming the early postoperative results (Figure 1c).

## 3. Discussion

Since its first description by Duckett in 1981, onlay preputial island flap urethroplasty has progressively achieved great recognition globally, and pediatric urologists dealing with these complex forms of hypospadias have early recognized the important advancement provided by this surgical technique [1,2,3,4,5,6].

Despite the improved surgical skills, the growing experience, and the body of information collected internationally on this surgical technique, the treatment of mid-shaft and proximal hypospadias continues to be a challenge for pediatric urologists even today, and perioperative complications remain a significant problem [3,4,5]. Even today, skin and glans dehiscence, urethral strictures and stenoses, and urethral fistulas are frequent short- and long-term postoperative complications following proximal and mid-shaft hypospadias repair, occurring in up to 30–68% of cases [3,4,6]. Various reasons for their occurrence have been identified, including inadequate blood supply, tension at the anastomosis, or damage of the intramural blood supply during urethroplasty [3,4]. 

Unsuccessful outcomes are mainly related to technical difficulties encountered during the surgical repair, since most children with severe hypospadias have a small penis, a narrow urethral plate, and a flat glans [3,4]. Therefore, in these patients, a preoperative course of androgens can be routinely prescribed to increase the penile length and circumference, and the glans width and vascularity, in this way facilitating the surgical repair [11,12,13,14,15]. Although preoperative androgens can increase the size of a poorly developed phallus, in the literature, several preclinical studies have shown that male sex steroid hormones can also prevent a normal cutaneous regenerative process or favor untoward bleedings. Hormone stimulation prior to hypospadias correction is, therefore, a controversial topic, and scarce scientific data can be found in the scientific literature to support such a practice [11,12,13,14,15]. 

New surgical techniques and intraoperative navigation systems are, therefore, required to help pediatric urologists guarantee optimal surgical outcomes in children with severe hypospadias. 

Aiming to reduce the rate of postoperative complications, nearly a hundred surgical methods have been spawned over time, and many effective techniques have been described more recently, including the modified transverse preputial island flap repair [16]. This technique involves the creation of a wedge or spoon anastomosis between the proximal urethral meatus and the neourethra, therefore providing support and blood supply for the neourethra. Additionally, it stretched the urethral plate width at the anastomosis and urethral meatus, effectively lowering the occurrence of postoperative urethral strictures [16].

Although many pediatric urologists prefer a two-stage procedure to guarantee long-lasting surgical outcomes, several studies have shown the high success of one-stage procedures in experienced hands. These are preferred by many surgeons to shorten the treatment period and reduce the economic and psychological burden frequently faced by children and their caregivers.

With the aim of further lowering the incidence of postoperative complications (especially those associated with a poor vascular supply), the development of innovative surgical techniques and intraoperative technologies has been highly encouraged internationally.

In this scenario, the administration of fluorescence dyes before or during surgery has progressively gained increasing popularity in many fields of adult and pediatric urology, where it has shown great potential to improve both surgical and functional outcomes, while minimizing anesthetic time and healthcare costs [7,8,9,10].

In this regard, ICG, a U.S. Food and Drug Administration (FDA)-approved water-soluble tricarbocyanine fluorophore, with a well-defined safety profile in humans, has been employed, with growing interest in several fields of pediatric urology, ranging from varicocele repairs to heminephrectomy, renal cyst deroofing, and renal tumor resection [7,8,9,10].

However, the clinical application of FGS in the field of hypospadias repair has never been described before [7,8,9,10]. Establishing adequate perfusion of the preputial island flap is a critical requisite for achieving long-lasting results, including a urethra of adequate size, and preventing postoperative complications, including urethral stenoses and fistulas. 

In this regard, identifying a proper vascular supply of the flap, and maintaining a good vascular pedicle (without tension or kink at the anastomosis site), are key elements for providing long-lasting anatomical and functional outcomes.

Our case report first described a step-by-step protocol to perform FGS during the surgical repair of a male infant with mid-shaft hypospadias. The EleVision IR system (Medtronic Ltd.) helped assess the vascular perfusion of the preputial island flap in real-time, and the performance of a procedure exempted from intra- and post-operative complications. 

In our hands, ICG-based FGS proved to be a safe and effective intraoperative imaging device. No ICG-related adverse effects occurred during or after surgery. We employed the 0.15 mg/kg dose and reached a high-quality fluorescence for several minutes, without experiencing any side effects. The dose we adopted was within the range reported in the literature to date (0.05 to 0.5 mg/kg) [7,8,9,10]. The vascular perfusion of the onlay preputial island flap was determined 80 s after the intraoperative administration of ICG. In comparison with the conventional clinical visual assessment of the tissue perfusion, ICG-based FGS offered objective data and significantly impacted intraoperative decision-making. In fact, it helped establish the proximal resection margin of the ventrally transposed prepuce. Furthermore, the EleVision IR system (Medtronic Ltd.) was also employed to monitor the perfusion of the ventral part of the neo-urethra. This step was considered important to support the vascular perfusion of the anastomosis and reduce the risk of postoperative complications in the neo-urethra, including dehiscence and stenosis, at this site.

In summary, the main advantages provided by this innovative intraoperative imaging modality were associated with excellent contrast, high sensitivity, and great spatial resolution of the preputial island flap, elevated on its pedicle. They were all provided without exposing the child to any ionizing radiation [7,8,9,10]. 

In the literature, several optical imaging devices have been described in children to detect the fluorescence signal and guide urological surgical procedures in real-time [7,8,9,10]. We opted for the EleVision IR system (Medtronic Ltd.). This intraoperative imaging device comes with two independent channels for the visible and the NIR signals, providing images of excellent quality, even with small doses of ICG.

Although further studies enrolling more patients undergoing hypospadias repair, and continued experience with fluorescent dyes and cameras, are needed to confirm our results and provide more robust data, we envisage that this technology has the potential to become part of the armory of pediatric hypospadiologists soon. The aim is to reduce the risk of intra- and postoperative complications associated with poor vascular supply, including skin and glans dehiscence, urethral stenosis, stricture, and fistulas. 

## 4. Conclusions

This report described the first step-by-step protocol for ICG-based laser angiography to guide the onlay preputial island flap urethroplasty in a child with mid-shaft hypospadias. The EleVision IR system (Medtronic Ltd.) proved to be safe, effective, and easy to employ, and should be considered as a reasonable adjunct for tissue perfusion assessment and operative decision-making in patients undergoing the onlay preputial island flap urethroplasty for severe hypospadias.

## Figures and Tables

**Figure 1 ijerph-20-06246-f001:**
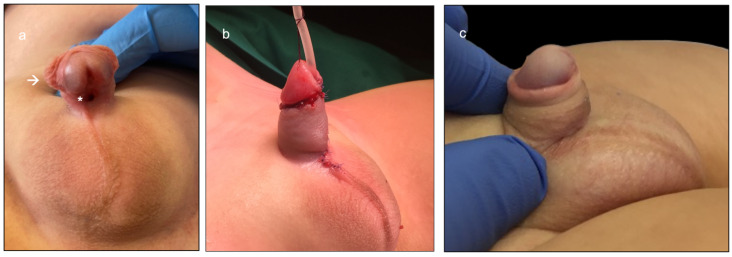
Aspect of the external genitalia of a 13-month-old boy referred to our tertiary referral hospital for mid-shaft hypospadias, who was scheduled for the onlay preputial island flap urethroplasty. (**a**) Preoperative aspect of the penis, characterized by a mid-shaft displacement of the external urinary meatus (*), mild ventral penile curvature, and a ventrally deficient hooded foreskin (→). (**b**) Immediate postoperative results, showing a straight penis with an apical neo-meatus. (**c**) Results at the 5-month follow-up, confirming the excellent esthetic and functional outcomes.

**Figure 2 ijerph-20-06246-f002:**
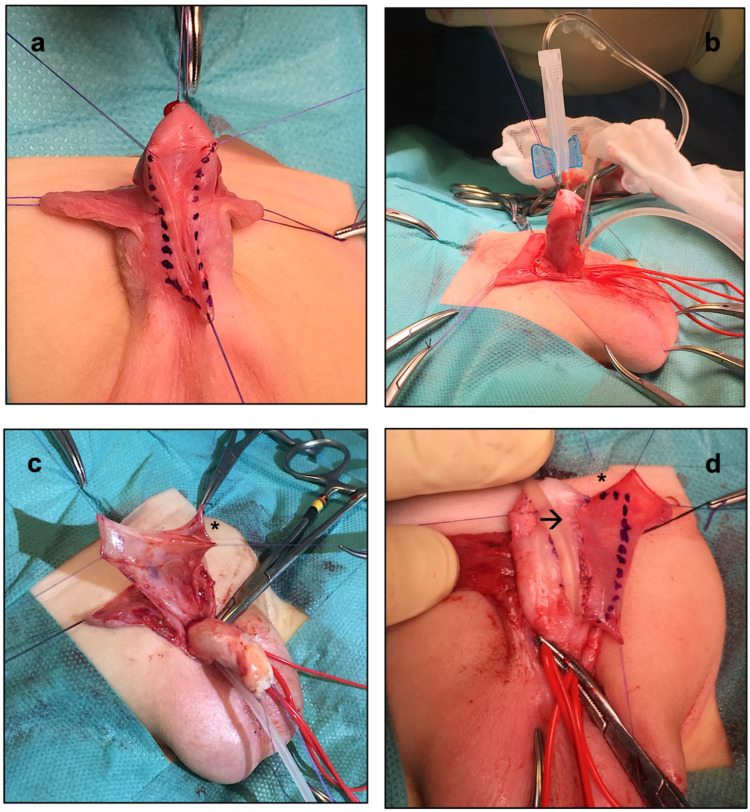
Main steps of the onlay preputial island flap urethroplasty. (**a**) Demarcation of the well-developed urethral plate. (**b**) Artificial erection test proving the lack of curvature after the release of the ventral skin chordees. (**c**) Transverse preputial island flap elevated on its pedicle (*). (**d**) Transverse preputial island flap (*) brought to the ventrum of the penis on its vascular pedicle sewn to the margins of the original urethral plate (→).

**Figure 3 ijerph-20-06246-f003:**
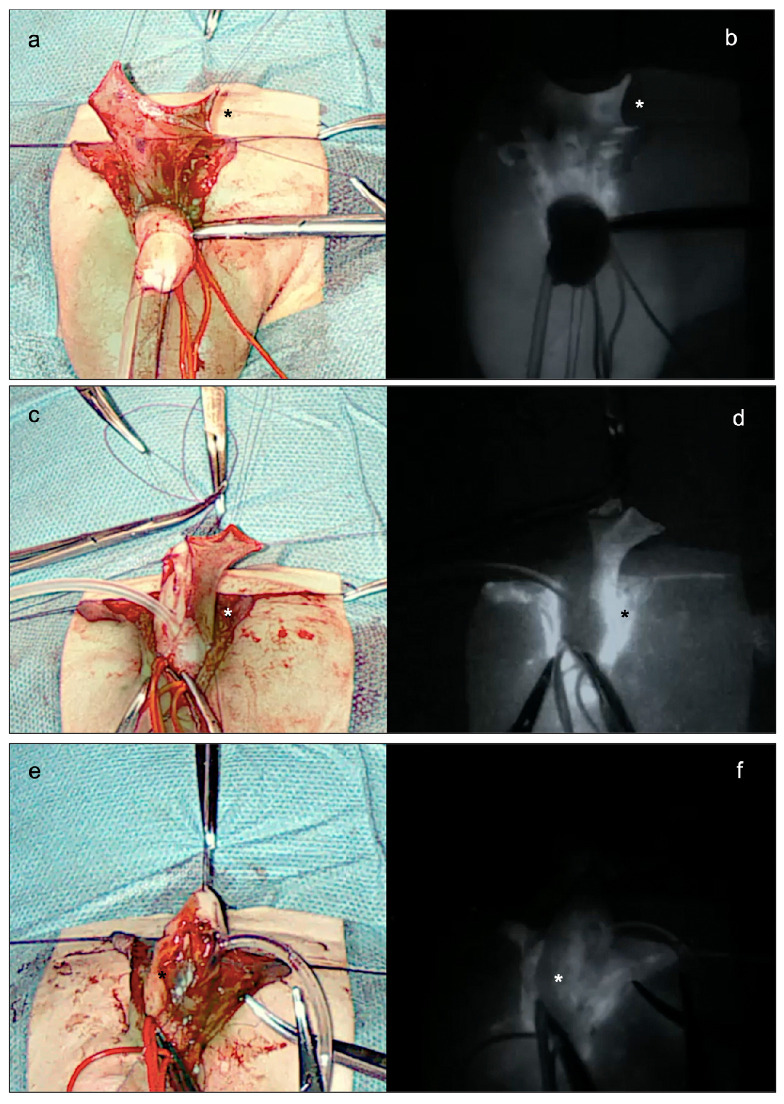
Main steps of the indocyanine green (ICG)-guided onlay preputial island flap urethroplasty in a 13-month-old boy referred to our tertiary referral hospital for a mid-shaft hypospadias. (**a,b**) Assessment of the vascular supply (*) of the transverse preputial island flap elevated on its pedicle, 80 s after the injection of 0.15 mg/kg of ICG (Diagnostic Green GmbH, Munich, Germany) and the incorporated the EleVision IR system (Medtronic Ltd.) on the surgical field. (**c,d**) Assessment of the vascular supply (*) of the transverse preputial island flap transposed ventrally and sewn to the margins of the urethral plate. (**e,f**) Assessment of the vascular supply (*) of the ventral part of the neo-urethra.

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
