# Peer review of "Indocyanine Green (ICG)-Guided Onlay Preputial Island Flap Urethroplasty for the Single-Stage Repair of Hypospadias in Children: A Case Report"

_ijerph, 2023, doi:10.3390/ijerph20136246_

Round 1
Reviewer 1 Report
Congratulations to the authors for this fine case report and thanks to the editor for giving me the chance to read and review this manuscript.
Case reports are important and needed for a better understanding of orphan diseases and for the development of new therapeutic strategies, especially in the surgical field.
The topic is highly interesting and young patients benefit from the development of new surgical strategies.
There are some points, the authors should highlight in their manuscript.
In my opinion it needs to be pointed out more clearly, that a sufficient blood supply of the pedicled skin graft is probably the most important factor for a successful surgery.
FGS can already be a helpful tool in the urologic field, as the authors mention it in the manuscript. Although it is legitimate to further develop the possibility of using FGS into the field of urologic reconstruction, like hypospadia repairs, they should provide more information about FGS in plastic surgery in general.
Since this is a case report, the authors have to give the reader more patient-information, especially intraoperative findings like the sufficiency of the urethral plate. How long was the urethral defect?
They mention that during artificial erection test “no residual ventral penile curvature” was observed. Why residual? Did they excise chordae? Did the patient receive androgen therapy prior to surgery?
There are several developments in the surgical techniques for proximal hypospadias repair recently. The authors should discuss more recent data and relate to them. (E.g.: DOI: 10.1007/s00345-023-04296-0).
The images presented are interesting but not easy to interpretate (maybe because of the size or the resolution). Maybe the authors can highlight the important features (e.g. by arrows).
The authors present a follow up of 5 months. In my opinion this is absolutely fine for a case report. But they have to mention, that strictures (a common complication) occur in a longer follow up and a longer fu is needed to make a sufficient statement about the efficacy of this procedure.
Thank you very much for the opportunity to read your work.
Author Response
Dear Reviewer,
Our group would like to thank you for the opportunity to resubmit a revised version of our manuscript.
Attached you can find a thoroughly revised version of the manuscript with the highlighted changes from the original version.
We thank you all very much for the useful comments, which have certainly improved the quality of our manuscript, and we hope it will now be acceptable for publication in IJERPH.
As requested, we have pointed out more clearly that a sufficient blood supply of the pedicled skin graft is one of the most important factors for a successful surgical repair.
Moreover, we have provided more patient-specific information, including the length of the urethral defect.
The adjective “residual” was added because, after the degloving of the penile skin, no more curvature was identified during the erection test. We have now clarified it in the manuscript. No androgen stimulation therapy was deemed required prior to surgery.
Furthermore, we have provided more recent data on new surgical techniques introduced for hypospadias repair, including the following study: “Lyu, Y., Chen, F., Xie, H. et al. One-stage repair of proximal hypospadias by in situ tubularization of the transverse preputial island flap. World J Urol 41, 813–819 (2023). https://doi.org/10.1007/s00345-023-04296-0”.
In addition, aiming to increase the interpretation of the images presented, we have added arrows and asterisks to the pictures to highlight the most important features. Moreover, a video has been added to the supplementary material.
Surgery was performed on the 18th of May 2022, so the postoperative follow-up has now increased to 11 months.
We look forward to your comments and suggestions regarding our re-submission.
Yours sincerely.
Reviewer 2 Report
In the title, the diagnosis or intervention of primary focus should be followed by the words “case report”.
In the introduction, the authors should state what is unique about this case and what does it add to the scientific literature?
Please state if there is adverse and unanticipated events.
Please submit the completed CARE checklist for your study.
Author Response
Dear Reviewer,
Our group would like to thank you for the opportunity to resubmit a revised version of our manuscript.
Attached you can find a thoroughly revised version of the manuscript with the highlighted changes from the original version, which has been written respecting the CARE checklist.
We thank you all very much for the useful comments, which have certainly improved the quality of our manuscript, and we hope it will now be acceptable for publication in IJERPH.
As requested, we have added the term “case report” in the title.
In the introduction, we have stated that the unicity of this case report relies on the fact that this is the first step-by-step protocol on how to perform fluorescence-guided surgery (FGS) in the repair of a male infant with a mid-shaft form of hypospadias.
As requested, we have stated that no adverse or unanticipated events have been experienced during and after surgery.
We look forward to your comments and suggestions regarding our re-submission.
Yours sincerely.
Reviewer 3 Report
The case study titled "Indocyanine green (ICG)-guided Onlay preputial island flap urethroplasty for the single-stage repair of hypospadias. The first experience." was well-written and well-presented.
Although there were a few grammatical errors, they did not detract from the overall quality of the study. However, the conclusion could benefit from being more elaborated.
Overall, the case study is in good shape and only requires minor revisions to address the English language and typo errors.
Author Response
Dear Reviewer,
Our group would like to thank you for the opportunity to resubmit a revised version of our manuscript.
Attached you can find a thoroughly revised version of the conclusion with the highlighted changes from the original version.
We thank you all very much for the useful comments, which have certainly improved the quality of our manuscript, and we hope it will now be acceptable for publication in IJERPH.
We look forward to your comments and suggestions regarding our re-submission.
Yours sincerely.
Reviewer 4 Report
were there any pre-clinical safety and efficacy studies done?
were ethics approval and patient consent obtained?
other options as comparison would be interesting
Author Response
Dear Reviewer,
Our group would like to thank you for the opportunity to resubmit a revised version of our manuscript.
Attached you can find a thoroughly revised version of the conclusion with the highlighted changes from the original version.
We thank you all very much for the useful comments, which have certainly improved the quality of our manuscript, and we hope it will now be acceptable for publication in IJERPH.
Indocyanine green (ICG) is a water-soluble fluorescent agent approved by the Food and Drug Administration (FDA) for use in various hepatic, cardiac, and ophthalmic clinical scenarios. Following intravenous administration, ICG rapidly binds plasma proteins and can be visualized using near-infrared fluorescence (NIRF). Urologists have used the agent “off-label” for a variety of clinical scenarios. In pediatric urology, ICG has been reported to aid in various renal perfusion applications during renal surgery and lymphography during varicocelectomy. Herein we report the use of ICG in hypospadias surgery.
No preclinical safety and efficacy studies are therefore required.
The ethics approval and patient consent were obtained for “off-label” uses.
We look forward to your comments and suggestions regarding our re-submission.
Yours sincerely.